# Synthesis and Evaluation of the Antifungal and Toxicological Activity of Nitrofuran Derivatives

**DOI:** 10.3390/pharmaceutics14030593

**Published:** 2022-03-08

**Authors:** Carolina Orlando Vaso, Fabiana Pandolfi, Níura Madalena Bila, Daniela De Vita, Martina Bortolami, Maria José Soares Mendes-Giannini, Valeria Tudino, Roberta Costi, Caroline Barcelos Costa-Orlandi, Ana Marisa Fusco-Almeida, Luigi Scipione

**Affiliations:** 1School of Pharmaceutical Science, Universidade Estadual Paulista, Araraquara 14800-903, SP, Brazil; carolovaso@hotmail.com (C.O.V.); niura.madalena.bila@gmail.com (N.M.B.); maria.giannini@unesp.br (M.J.S.M.-G.); carolbarceloscosta@gmail.com (C.B.C.-O.); 2Department of Scienze di Base e Applicate per l’Ingegneria, Sapienza University of Rome, Via Castro Laurenziano 7, 00185 Rome, Italy; fabiana.pandolfi@uniroma1.it (F.P.); martina.bortolami@uniroma1.it (M.B.); 3Department of Environmental Biology, Sapienza University of Rome, Piazzale Aldo Moro 5, 00185 Rome, Italy; daniela.devita@uniroma1.it; 4Department of Chimica e Tecnologia del Farmaco, Sapienza University of Rome, Piazzale Aldo Moro 5, 00185 Rome, Italy; valeria.tudino@uniroma1.it; 5Department of Chemistry and Technology of Drug, Instituto Pasteur, Fondazione Cenci Bolognetti, Sapienza University of Rome, Piazzale Aldo Moro 5, 00185 Rome, Italy; roberta.costi@uniroma1.it

**Keywords:** broad-spectrum antifungal, nitrofuran derivates, antifungal activity, *Candida* sp., *Cryptococcus neoformans*, *Histoplasma capsulatum*, *Paracoccidioides brasiliensis*, *Trichophyton rubrum*, *Trichophyton mentagrophytes*, *Caenorhabditis elegans* larvae

## Abstract

Fungal diseases affect more than 1 billion people worldwide. The constant global changes, the advent of new pandemics, and chronic diseases favor the diffusion of fungal pathogens such as *Candida*, *Cryptococcus*, *Aspergillus*, *Trichophyton*, *Histoplasma capsulatum,* and *Paracoccidioides brasiliensis*. In this work, a series of nitrofuran derivatives were synthesized and tested against different fungal species; most of them showed inhibitory activity, fungicide, and fungistatic profile. The minimal inhibitory concentration (MIC_90_) values for the most potent compounds range from 0.48 µg/mL against *H. capsulatum* (compound **11**) and *P. brasiliensis* (compounds **3** and **9**) to 0.98 µg/mL against *Trichophyton rubrum* and *T. mentagrophytes* (compounds **8**, **9**, **12**, **13** and **8**, **12**, **13**, respectively), and 3.9 µg/mL against *Candida* and *Cryptococcus neoformans* strains (compounds **1** and **5**, respectively). In addition, all compounds showed low toxicity when tested in vitro on lung cell lines (A549 and MRC-5) and in vivo in *Caenorhabditis elegans* larvae. Many of them showed high selectivity index values. Thus, these studied nitrofuran derivatives proved to be potent against different fungal species, characterized by low toxicity and high selectivity; for these reasons, they may become promising compounds for the treatment of mycoses.

## 1. Introduction

Fungal diseases annually affect more than 1 billion people worldwide. These infections present a wide variety of symptoms, but they can often become invasive, especially in immunocompromised patients, with the risk of leading to death. Their mortality rate is highly relevant, as it represents more than 1.6 million deaths annually, similar to the tuberculosis mortality rate and three times higher than that recorded for malaria. Fungal diseases have increased due to the number of susceptible individuals, including people living with the human immunodeficiency virus (HIV), hematopoietic stem cell or organ transplant recipients, patients with malignancies, or immunological conditions receiving immunosuppressive treatment, premature neonates, and the elderly. More recently, the SARS-CoV-2 (COVID-19) virus pandemic has been associated with some cases of fungal diseases in hospitalized patients [1,2,3,4,5].

Fungal pathogens such as *Candida* and *Cryptococcus* are distributed worldwide and constitute the majority of invasive fungal infections (IFIs). Dimorphic fungi, such as *Histoplasma capsulatum* and *Paracoccidioides* spp., are geographically restricted to their respective habitats and cause endemic mycoses. Dermatophytosis is globally considered the most common dermatological disease [6].

Among these fungal diseases, candidiasis, caused by yeasts of the *Candida* genus, commonly affects the gastrointestinal tract, urinary tract, and oral cavity or becomes systemic, affecting the entire organism [7]. The annual incidence of candidiasis is around 2 million cases, and the disseminated figure is around 700,000 cases [1]. *Candida albicans* is the most prevalent species; however, lately, non-*albicans* species have increased significantly. The ascendant species are *C. parapsilosis*, *C. tropicalis*, *C. krusei*, *C. glabrata*, *C. guilliermondii* [6,7,8,9], as well as *C. auris*. *Candida* species infections are usually treated with drugs from the class of polyenes (amphotericin B and nystatin), and azoles, such as fluconazole, clotrimazole, and miconazole [7]. However, reports of fungal resistance have already been described for the standard antifungal classes used in therapy. Regarding polyenes, mutations change the principal sterol in the membrane, affecting the polyene binding, whereas, for the azole class, mutations in the gene encoding the target protein, or its overexpression, as well as other resistance mechanisms can occur [10,11,12]. In addition, reports of high toxicity and drug interactions are prevalent and have been described for these antifungal classes [13,14,15].

Encapsulated yeasts represent 200,000 cases of fungal diseases annually, mainly in individuals with the human immunodeficiency virus (HIV) [16]. The main species is *Cryptococcus neoformans* [17,18], capable of causing severe meningoencephalitis [19,20]. The most common treatment available is based on the application of polyenes, azoles, and flucytosine, depending on the severity and immunological status of the host [21,22]. When the fungus reaches the central nervous system, the main challenge in the treatment is for the drug to cross the blood-brain barrier. Amphotericin B is the most commonly indicated treatment; however, it requires hospitalization and the monitoring of liver and kidney function [21]. Cases of antifungal resistance have been reported for *Cryptococcus* sp. and reports of host toxicity, which are aggravated by a long period of treatment [21,23,24].

*Histoplasma capsulatum* and *Paracoccidioides* sp. cause respiratory illnesses that can become widespread [25,26]. The annual incidence of histoplasmosis is around 500,000 cases. It is a mycosis considered to have a worldwide distribution, already having been described in all continents except for Antarctica [1,27]. Paracoccidioidomycosis is an endemic infection in Latin America, and most cases occur in Brazil, Argentina, Colombia, Ecuador, Venezuela, and Paraguay [28,29]. The standard treatment for both mycoses is amphotericin B and itraconazole; however, the antibacterial cotrimoxazole (sulfamethoxazole/trimethoprim combination) is also indicated for paracoccidioidomycosis [30,31,32,33]. Cases related to nephrotoxicity, hepatotoxicity, and drug interactions increase considerably in these diseases, as they are considered systemic and require long treatments that can last for up to 24 months [26,31,33,34].

Dermatophytosis is caused by filamentous fungi that have a predilection for keratin. They mainly affect fur, hair, nails, and skin [35,36]. These mycoses affect about 20 to 25% of the world’s human population [8,37]. *Trichophyton rubrum* and *T. mentagrophytes* are the most prevalent [37,38]. The usual treatment is carried out with drugs from the family of azoles and allylamines. However, prolonged treatment time, recurrent infections, and frequent reports of resistant strains to conventional drugs are the main limitations in the treatment of these mycoses [38,39,40].

The search for new molecules capable of effectively treating fungal infections and causing minimal toxicity is constant. Among these molecules, nitrofurans are compounds with a 5-nitrofuran ring and different substituents in position 2. The first nitrofuran was described in 1944, and these drugs were widely used for decades in the field of agriculture to prevent and control diseases and were added to animal feed to stimulate growth [41,42]. In the 1990s, Europe banned the use of nitrofurans for agricultural purposes, and later in 2002, other countries such as the United States and China also banned them for animal use, due to the residues that these drugs left behind in the meat. These residues possibly cause side effects in human beings, such as hematological abnormalities (aplastic anemia), in addition to their carcinogenic, mutagenic, and genotoxic effects. Currently, many research groups have been studying nitrofuran formulations, trying to reduce their toxicity and side effects, as they have excellent antimicrobial activity [41,43,44].

There are few reports in the literature on the study of the antifungal activity of nitrofurans; some authors have shown a potent antifungal and anti-biofilm activity against *Candida* species, with the capacity to inhibit cell adhesion and aggregation [45,46]. Regarding the toxicity assessment, some studies with nitrofuran derivatives have shown that these compounds have low toxicity when tested on human cell lines [45,46,47].

In this study, we synthesized a series of ester, amide, or chalcone 5-nitrofuran derivatives, as depicted in Figure 1, intending to assess their antifungal activity against a wide range of fungal species and to evaluate their toxic effects on human cells in vitro and on *Caenorhabditis elegans* larvae in vivo.

## 2. Materials and Methods

### 2.1. Chemical Synthesis

The compounds **1**, **4–14** were synthesized according to the literature (**1** [45], **4** [48], **5** [49], **6** [50], **7** [51], **8–12** [52], **13–14** [53]), and all the analytical data were in accordance with those reported previously (Appendix A). The compounds **2**, **3** [54], and **15–17** [55,56] were not previously described and were synthesized following literature methods; the detailed chemical procedures and the related spectroscopic data are reported below. All reagents and solvents were of analytical grade and were purchased from Sigma-Aldrich (Milano, Italy) or from Fluorochem (Hadfield, UK). Column chromatographies were performed on silica gel (Merck; 63−200 μm particle size). ^1^H NMR and ^13^C NMR spectra were acquired at 25 °C, unless otherwise specified, on a Bruker AVANCE-400 spectrometer at 9.4 T (Bruker, Billerica, MA, USA), operating at 400 MHz (^1^H NMR) and 100 MHz (^13^C-NMR); chemical shift values (δ) are given in ppm relative to TMS, using the solvent as the internal reference, while coupling constants are given in Hz. The following abbreviations were used: s = singlet, d = doublet, t = triplet, dd = double doublet, dt = double triplet, and m = multiplet. Mass spectra were recorded on a ThermoFinnigan (San Jose, CA, USA) LCQ Classic LC/MS/MS ion trap, equipped with an ESI source and a syringe pump; samples (10^−4^–10^−5^ M in MeOH/H_2_O 80:20) were infused in the electrospray system at a flow rate of 5−10 μLmin^−1^; when necessary, 50 μL of 10^−2^ M HCOOH was added to the sample solutions to promote analyte ionization; the ESI-MS data are given as *m*/*z*, with mass expressed in amu. Melting points were determined on a FALC Mod. 360 D (Falc Instruments, Treviglio, Italy) or on a Kofler apparatus and are uncorrected. Infrared spectra were recorded on a PerkinElmer (Waltham, MA, USA) Spectrum One FT-IR spectrometer in a nujol mull. The purity of the compounds was determined by elemental analyses, obtained by a PE 2400 (PerkinElmer, Waltham, MA, USA) analyzer, and the analytical results were within ±0.4% of the theoretical values for all compounds.

#### 2.1.1. General Procedure for the Synthesis of Compounds *N*-(3-(1*H*-imidazol-1-yl)propyl)-5-nitrofuran-2-carboxamide (**2**) and 5-nitro-*N*-(2-(pyridin-2-yl)ethyl)furan-2-carboxamide (**3**)

As reported in Figure 1, the 5-nitrofuran-2-carboxylic acid was added to a suspension of 1,1-carbonyldiimidazole (CDI) in 1,4-dioxane in a 1:1 molar ratio, and the reaction mixture was stirred at rT for 2 h. Then, the opportune amine (1 eq) was added, and the mixture was stirred at rT for 12 h and refluxed for an additional 2 h. After this time had passed, the reaction mixture was treated with 2 mL of H_2_O and refluxed for 1 h. The solvent was removed under reduced pressure and the residue was treated with CH_2_Cl_2_ (10 mL) and NaOH (10 mL, 1 N). The organic phase was separated and washed with 10 mL of H_2_O, dried over anhydrous Na_2_SO_4_, then filtered and concentrated in a vacuum. The obtained residue was subjected to silica gel column chromatography using AcOEt/MeOH as an eluent to afford the purified amide compounds **2** and **3**.

##### *N*-(3-(1*H*-imidazol-1-yl)propyl)-5-nitrofuran-2-carboxamide (**2**)

Compound **2** was prepared using 5-nitrofuran-2-carboxylic acid (785 mg, 5 mmol), CDI (811 mg, 5 mmol) and 3-(1*H*-imidazol-1-yl)propan-1-amine (597 µL, d = 1.049 g/mL, 5 mmol) in 25 mL of 1,4-dioxane, following the general procedure. Compound **2** was obtained with 7% yield; R_f_ = 0.59 (AcOEt/MeOH 9:1). mp: 144–148 °C (Kofler). ESI-MS (*m*/*z*): 265.4 [M + H]^+^. Anal. (C_11_H_12_N_4_O_4_) C, H, N; calcd: C 50.00%, H 4.58%, N 21.20%; found: C 50.08%, H 4.58%, N 21.18%. IR (nujol mull, cm^−^^1^): 3107; 1657; 1587; 1464; 1284. ^1^H-NMR (MeOD) δ (ppm): 7.71 (s, 1H); 7.55 (d, 1H, *J* = 3.6 Hz); 7.30 (d, 1H, *J* = 3.6 Hz); 7.19 (s, 1H); 6.98 (s, 1H); 4.13 (t, 2H, *J* = 6.8 Hz); 3.42 (t, 2H, *J* = 6.8 Hz); 2.12 (m, 2H). ^13^C-NMR (MeOD) δ (ppm): 157.5; 151.8; 147.9; 137.1; 127.7; 119.2; 115.5; 111.9; 44.2; 36.3; 30.4 (Appendix A).

##### 5-nitro-*N*-(2-(pyridin-2-yl)ethyl)furan-2-carboxamide (**3**)

Compound **3** was prepared using 5-nitrofuran-2-carboxylic acid (204 mg, 1.3 mmol), CDI (211 mg, 1.3 mmol) and 2-(pyridin-2-yl)ethanamine (156 µL, d = 1.021 g/mL, 1.3 mmol) in 9 mL of 1,4-dioxane, following the general procedure. Compound **3** was obtained with 30% yield. mp: 52–54 °C (Kofler). ESI-MS (*m*/*z*): 265.4 [M + H]^+^. Anal. (C_12_H_11_N_3_O_4_) C, H, N; calcd: C 55.17%, H 4.24%, N 16.09%; found: C 55.23%, H 4.25%, N 16.06%. IR (nujol mull, cm^−^^1^): 3297; 1653; 1583; 1465; 1299. ^1^H-NMR (MeOD) δ (ppm): 8.48 (d, 1H, *J* = 4.8 Hz); 7.77 (dt, 1H, *J* = 1.8 Hz, 7.8 Hz); 7.52 (d, 1H, *J* = 3.8 Hz); 7.37 (d, 1H, *J* = 7.6 Hz); 7.26–7.31 (m, 2H); 3.75 (t, 2H, *J* = 7.2 Hz); 3.10 (t, 2H, *J* = 7.2 Hz). ^13^C-NMR (MeOD) δ (ppm): 158.6; 148.5; 148.0; 147.3; 143.1; 137.4; 123.7; 121.9; 115.3; 111.8; 39.1; 36.7 (Appendix A).

#### 2.1.2. General Procedure for the Synthesis of Compounds (*E*)-1-(4-(methylsulfonyl)phenyl)-3-(5-nitrofuran-2-yl)prop-2-en-1-one (**15**), (*E*)-1-(2,4-dichlorophenyl)-3-(5-nitrofuran-2-yl)prop-2-en-1-one (**16**) and (*E*)-1-(2,4-dichloro-5-fluorophenyl)-3-(5-nitrofuran-2-yl)prop-2-en-1-one (**17**)

The 5-nitrofuran-2-carbaldehyde and the opportune acetophenone, in a 1:1 molar ratio, were dissolved in 1.68 mL of acetic acid and sulfuric acid (67 μL, 98%) was added to the solution. The reaction mixture was stirred at 100 °C for 24 h (Figure 2). After this time, the mixture was extracted with CH_2_Cl_2_ (3 × 25 mL). The organic phase was dried over anhydrous Na_2_SO_4_, filtered, and concentrated in a vacuum. The obtained residue was subjected to silica gel column chromatography to afford the purified compounds **15–17**.

##### (*E*)-1-(4-(methylsulfonyl)phenyl)-3-(5-nitrofuran-2-yl)prop-2-en-1-one (**15**)

Compound **15** was prepared using 5-nitrofuran-2-carbaldehyde (105 µL, d = 1.349 g/mL, 1 mmol) and 1-(4-(methylsulfonyl)phenyl)ethanone (198 mg, 1 mmol), following the general procedure. Compound **15** was obtained as a yellow solid with 75% yield; R_f_ = 0.59 (CH_2_Cl_2_/AcOEt 9.5:0.5). mp: 197–200 °C (dec). ESI-MS (*m*/*z*): 322.1 [M + H]^+^. Anal. (C_14_H_11_NO_6_S) C, H, N; calcd: C 52.33%, H 3.45%, N 4.36%; found: C 52.24%, H 3.44%, N 4.37%. IR (nujol mull, cm^−^^1^): 3127; 3096; 1664; 1608; 1561; 1465; 1346. ^1^H-NMR (DMSO-*d_6_*) δ (ppm): 8.32 (d, 2H, *J* = 7.9 Hz); 8.12 (d, 2H, *J* = 7.8 Hz); 7.89–7.81 (m, 2H); 0.64 (d, 1H, *J* = 15.7 Hz); 7.47 (d, 1H, *J* = 3.9 Hz); 3.30 (s, 3H). ^13^C-NMR (DMSO-*d_6_*) δ (ppm): 188.4; 153.4; 152.6; 145.0; 141.0; 130.4; 129.9; 128.0; 125.2; 118.9; 115.2; 43.7 (Appendix A).

##### (*E*)-1-(2,4-dichlorophenyl)-3-(5-nitrofuran-2-yl)prop-2-en-1-one (**16**)

Compound **16** was prepared using 5-nitrofuran-2-carbaldehyde (105 µL, d = 1.349 g/mL, 1 mmol) and 1-(2,4-dichlorophenyl)ethanone (189 mg, 1 mmol), following the general procedure. Compound **16** was obtained as a yellow solid with 17% yield; R_f_ = 0.64 (CH_2_Cl_2_/AcOEt 9.5:0.5). mp: 164–165 °C. Anal. (C_13_H_7_Cl_2_NO_4_) C, H, N; calcd: C 50.03%, H 2.26%, N 4.49%; found: C 50.18%, H 2.26%, N 4.50%. IR (nujol mull, cm^−^^1^): 3107; 1657; 1587; 1548; 1464; 1284. ^1^H-NMR (Acetone-*d_6_*) δ (ppm): 7.67–7.64 (m, 2H); 7.61 (d, 1H, *J* = 4.0 Hz); 7.56 (dd, 1H, *J* = 2.0 Hz, *J* = 8.4 Hz); 7.44 (d, 1H, *J* = 16.0 Hz); 7.29–7.25 (m, 2H). ^13^C-NMR (Acetone-*d_6_*) δ (ppm): 190.7; 152.8; 152.5; 137.0; 136.8; 132.0; 131.0; 130.0; 129.9; 128.1; 127.7; 118.0; 113.5 (Appendix A).

##### (*E*)-1-(2,4-dichloro-5-fluorophenyl)-3-(5-nitrofuran-2-yl)prop-2-en-1-one (**17**)

Compound **17** was prepared using 5-nitrofuran-2-carbaldehyde (105 µL, d = 1.349 g/mL, 1 mmol) and 1-(2,4-dichloro-5-fluorophenyl)ethanone (145 µL, d = 1.425 g/mL, 1 mmol), following the general procedure. Compound **17** was obtained as a yellow solid with 30% yield; R_f_ = 0.52 (CH_2_Cl_2_/Hexane 7:3). mp: 163–165 °C. Anal. (C_13_H_6_Cl_2_FNO_4_) C, H, N Calcd: C 47.30%, H 1.83%, N 4.24%; Found: C 48.01%, H 1.83%, N 4.25%. IR (nujol mull, cm^−^^1^): 3089; 3070; 1660; 1612; 1463; 1346. ^1^H-NMR (DMSO-*d_6_*) δ (ppm): 8.01 (d, 1H, *J* = 6.4 Hz); 7.83–7.79 (m, 2H); 7.46–7.40 (m, 2H); 7.17 (d, 1H, *J* = 16.0 Hz). ^13^C-NMR (DMSO-*d_6_*) δ (ppm): 190.3; 156.1 (d, *J* = 249.2 Hz); 152.4; 152.2; 137.8 (d, *J* = 6.1 Hz); 131.9; 131.6; 127.8; 126.1 (d, *J* = 3.7 Hz); 122.9 (d, *J* = 19.0 Hz); 119.0; 117.7 (d, *J* = 24.3 Hz); 114.6 (Appendix A).

### 2.2. Antifungal Drugs and Nitrofuran Derivatives

The antifungal drugs amphotericin B (AmB) and terbinafine (TRB) were purchased commercially (Sigma-Aldrich, Milano, Italy).

The nitrofuran derivatives were solubilized in 100% dimethyl sulfoxide (DMSO) at a stock concentration of 30,000 µg/mL and were stored at −80 °C. Antifungal drugs stock solutions were prepared considering their purity, using the calculations recommended in document M27-A3. The working solutions of the compounds (0.06–250 µg/mL) and the drugs AmB (Sigma-Aldrich, Milano, Italy) (0.007–4 µg/mL) and TRB (Sigma-Aldrich, Milano, Italy) (0.001–1 µg/mL) were prepared in Roswell Park Memorial Institute (RPMI)-1640 medium with L-glutamine, without sodium bicarbonate, and with phenol red as the pH indicator (Gibco^®^ -Thermo-Fisher-Scientific, Waltham, MA, USA), buffered with 4-morpholinepropanesulfonic acid hemisodium salt (MOPS) (Sigma-Aldrich, Milano, Italy) with 2% glucose (Synth, Diadema, São Paulo, Brazil) pH = 7.

### 2.3. Microorganisms and Culture Conditions

The following fungi (strains) were used: *Candida albicans* (ATCC 90028); *C. krusei* (ATCC 6258); *C. glabrata* (ATCC 90030); *Cryptococcus neoformans* (H99/ATCC 208821); *Histoplasma capsulatum* (G217-B/ATCC 26032), *Paracoccidioides brasiliensis* (Pb 18), originally isolated from a case of pulmonary paracoccidioidomycosis in São Paulo, SP, Brazil [57], *Trichophyton mentagrophytes* (ATCC 11481), and *T. rubrum* (ATCC 28289). The strains of *Candida* and *Cryptococcus* were subcultured on Sabouraud dextrose agar (BD Difco™, Wokingham, Berkshire, UK) for 24 and 48 h, respectively, as described in document M27-A3 of the Clinical Institute Standards Laboratory (CLSI, Wayne, PA, USA) [58]. For *H. capsulatum*, the yeast phase of each strain was maintained in Brain and Heart Infusion (BHI) agar (BD Difco™, Wokingham, Berkshire, UK), supplemented with 0.1% L-cysteine (Synth, Diadema, São Paulo, Brazil) and 1% glucose (Sigma-Aldrich, Milano, Italy) for 96 h at 37 °C. The strains were subsequently subcultured in Ham’s F-12 Nutrient Mixture medium (HAM-F12) (Gibco^®^ Thermo-Fisher-Scientific, Waltham, MA, USA) supplemented with 1.8% glucose (Synth, Diadema, São Paulo, Brazil), 0.1% glutamic acid (Synth, Diadema, São Paulo, Brazil), 0.6% HEPES (Sigma-Aldrich, Milano, Italy) and 0.0008% L-cysteine (Synth, Diadema, São Paulo, Brazil) at 37 °C, for 96 h and with shaking at 150 rpm [59,60]. *P. brasiliensis* was kept in Fava-Netto medium at 37 °C for 96 h [61]. The strains of *Trichophyton* were kept in malt extract agar (malt extract (Kasvi, São José do Pinhais, Paraná, Brazil): 2%; peptone from animal tissue (Sigma-Aldrich, Milano, Italy): 2%; glucose (Synth, Diadema, São Paulo, Brazil): 2%; agar (Kasvi, São José do Pinhais, Paraná, Brazil: 2%), pH 5.7, and incubated at 28 °C for 7 days or until sporulation [35,38].

### 2.4. Fungal Susceptibility to Nitrofuran Derivates and Antifungal Drugs

#### 2.4.1. *Candida* sp. and *Cryptococcus neoformans*

The susceptibility test for *Candida* and *C. neoformans* species was conducted as recommended in CLSI M27-A3 [58]; yeasts were adjusted at a concentration of 5 × 10^6^ cells/mL, then a dilution of 1:50 was performed using 0.85% NaCl and 1:20 using RPMI-1640 medium (Gibco^®^ -Thermo-Fisher-Scientific, Waltham, MA, USA). Dilutions of the compounds and antifungal reference drugs were dispensed in a 96-well microplate (Kasvi, São José do Pinhais, Paraná, Brazil) at a total volume of 100 µL/well; subsequently, 100 µL/well of the inoculum was added and the plates were incubated at 37 °C for 24 h (*Candida* sp.) and 48 h (*C. neoformans*). A visual and colorimetric readout was performed with 0.03% resazurin (Sigma-Aldrich, Milano, Italy) [38,61,62]. Quality control was performed with *C. krusei* ATCC 6258 strains, using the drug AmB. MIC was considered when inhibition was at least 90% of the growth when compared to the control (MIC_90_).

#### 2.4.2. *Histoplasma capsulatum*

Susceptibility assays for *H. capsulatum* were performed according to the document M27-A3, proposed by CLSI [58], with modifications as proposed by Li and collaborators [63], Wheat and collaborators [64], and Kathuria and collaborators [65]. The inoculum was prepared in 0.85% NaCl, then the cell viability was checked with a hemocytometer using Trypan blue (Gibco^®^ Thermo-Fisher-Scientific, Waltham, MA, USA) in a 1:1 ratio. The fungal suspension yeasts were adjusted at a 5 × 10^6^ cells/mL concentration in 0.85% NaCl. Then, 1:10 dilution was performed and the fungal suspensions were placed in contact with the serial dilutions of the compounds and reference drug. The final fungal concentration was 2.5 × 10^5^ cells/mL. The plates were incubated for 144 h at 37 °C while shaking at 150 rpm. Visual and colorimetric readings were performed by adding 30 µL of 0.03% resazurin.

#### 2.4.3. *Paracoccidioides brasiliensis*

The susceptibility was conducted as described by de Paula e Silva et al. [61]; fungal suspensions were prepared at a 5 × 10^6^ cells/mL concentration, diluted to 1:50 in 0.85% NaCl and 1:20 in RPMI-1640 medium. Serial dilution of the compounds was carried out and the result was placed in contact with the fungal suspension. The plates were incubated at 37 °C for 48 h at 150 rpm, and 20 µL of Alamar Blue (Invitrogen- Thermo-Fisher-Scientific, Waltham, MA, USA) was added and incubated for another 24 h for colorimetric readings.

#### 2.4.4. *Trichophyton rubrum* and *T. mentagrophytes*

The experiments were performed according to CLSI M38-A2 [65] for dermatophytes, with minor modifications as described by Costa-Orlandi et al. [35]. The fungal suspensions were adjusted by counting the conidia in the hematocytometer to reach a final concentration of 2.5 × 10^3^ cells/mL in RPMI-1640 medium. This was added to the wells at the appropriate compound concentrations and incubated at 35 °C for 96 h with agitation (150 rpm). Visual and colorimetric readings were taken with the addition of 20 µL of 0.03% resazurin [61]. Quality control was performed using the strain of *T. rubrum* ATCC MYA-4438, with the drug TRB.

### 2.5. Determination of Minimum Fungicide Concentration (MFC)

The minimum fungicidal concentration was performed as described by Costa-Orlandi et al. [35]. To be precise, 100 µL aliquots of the contents of the wells were removed and plated in specific media for each fungus. *Candida* spp. and *C. neoformas* were plated on Sabouraud dextrose agar (BD Difco™) and incubated at 37 °C for 24 h and 48 h, respectively. *H. capsulatum* was plated on BHI agar supplemented with 0.1% L-cysteine and 1% glucose and was subsequently incubated at 37 °C for 96 h; *P. brasiliensis* was plated on Fava-Neto medium at 37 °C for 72 h and dermatophytes on Sabouraud agar at 28 °C for 96 h. Concentrations greater than or equal to MIC_90_ were used. The minimum fungicidal concentration is the lowest concentration of the compound or drug where the development of 99.9% of microorganisms does not occur [35].

### 2.6. Cell Line Maintenance

Two cell lines were used: the A549 (ATCC^®^ CCL-185) lung epithelial cell line and the MRC-5 (ATCC^®^ CCL-171) pulmonary fibroblast cell line. Both were grown in Dulbecco’s Modified Eagle’s Medium (DMEM) (Gibco), supplemented with 10% fetal bovine serum (FBS) (Sigma-Aldrich, Milano, Italy), and incubated in 5% CO_2_ at 37 °C [66]. After thawing, the cell lines were expanded to reach 80% confluence, ready to be trypsinized and transferred to another bottle.

#### Cytotoxicity Assay in Monolayer Models by Resazurin Colorimetric Method

Cytotoxicity tests were performed for both strains (A549 and MRC-5) to verify the selectivity index (SI). The assay was performed as described by Costa-Orlandi et al. [35] and Bila et al. [38]. Cell suspensions were prepared to obtain a final concentration of 2 × 10^4^ cells/well in a 96-well microdilution plate. After 24 h of incubation, the culture medium was removed, and 200 μL of different concentrations of nitrofuran derivates were added. All the plates were incubated for 72 h. After incubation, 20 μL of resazurin (Sigma-Aldrich, Milano, Italy) at 60 μM was added, and the plates were further incubated for 8 h. Cell viability was assessed based on spectrophotometric analysis (Epoch, Biotek, Santa Clara, CA, USA) at wavelengths of 570 nm and 600 nm [67]. The SI was calculated according to the ratio between the IC_50 _ and the MIC_90_ [68].

### 2.7. Toxicity In Vivo on C. elegans Model

The experiments were carried out with the mutant strain (AU37 [glp-4 (bn2) I; sek-1 (km4) X]). The strain was maintained on plates containing nematode growth medium (NGM) seeded with *Escherichia coli* OP50 at 16 °C. The stage of synchronization of young adults at stage L4 was performed for the toxicity test. About 20 larvae were transferred to each well of 96-well plates containing 100 µL of a medium composed of 60% of 50 mM NaCl; 40% BHI broth; 10 mg/mL cholesterol in ethanol; 200 mg/mL ampicillin and 90 mg/mL kanamycin. Then, 100 μL of dilutions of nitrofuran derivatives (showing the best selectivity index against the fungal species) were added. Final concentrations in each well ranged from 250 to 31.25 μg/mL. Plates were incubated at 25 °C for 24 h. Survival was assessed by the mobility and shape of the nematode (stick-shaped larvae were considered dead, while sinusoidal larvae were considered alive) under an inverted optical microscope on 40× objective lenses [35,69,70,71].

### 2.8. Statistic Analysis

All tests were carried out in triplicate and in three independent experiments. Data were subjected to statistical analysis using an analysis of variance (one-way ANOVA) with a Bonferroni post-test, using GraphPad Prism 5.0 software (GraphPad Software Inc., La Jolla, CA, USA). All *p* values of less than 0.05 were considered statistically significant.

## 3. Results

### 3.1. Chemical Synthesis

The synthesis of 5-nitrofuran derivatives **2** and **3** involves the activation of the 5-nitro-furan-2 carboxylic acid with carbonyl-diimidazole (CDI) and the subsequent reaction with the appropriate amine. The obtained amide derivatives were purified by silica gel column chromatography and characterized by MS-ESI spectrometry, IR, ^1^H and ^13^C-NMR spectroscopy; the analytical data were in accordance with the proposed structures. Specifically, in addition to the expected molecular peaks in the ESI-MS spectra, the shifts of the methylene signals, from 2.6 to 3.4 ppm for compound **2** and from about 2.9 to 3.7 ppm for compound **3** due to the transformation from primary amine to amide, are diagnostic. Furthermore, the presence of carbonyl stretching at 1657 cm^−^^1^ and 1650 cm^−^^1^ for **2** and **3**, respectively, indicate the extension of conjugation and further confirm the proposed structures.

For the synthesis of nitrofurans **15–17**, the 5-nitrofuran-2-carbaldehyde reacted with the opportune acetophenone in the presence of sulfuric and acetic acid. The obtained chalcones were purified by silica gel column chromatography and characterized by spectroscopic techniques. In particular, the low-frequency shift of the carbonyl stretching signal observed in the chalcones, compared to the corresponding acetophenones, supports the double-bond formation (1664 vs. 1685 cm^−^^1^ for **15**, 1677 vs. 1698 cm^−^^1^ for **16**, and 1660 vs. 1707 cm^−^^1^ for **17**); these data are confirmed by ^13^C-NMR spectra, which show the presence of vinyl carbons. The expected E geometry of the double bond is confirmed by the typical ^3^J values of approximately 16 Hz of the trans vinyl coupling constant observed in the ^1^H-NMR spectra of compounds **15–17**.

### 3.2. Broad-Spectrum Antifungal Activity

In general, yeasts were resistant to most compounds with a prevalence of MIC_90_ greater than or equal to 250 µg/mL in 72% of the compounds against *Candida* species and 47% against *C. neoformans* (Figure 2). *H. capsulatum* (41%), *P. brasiliensis* (47%), and dermatophytes (35%) were more sensitive to nitrofurans, with a prevalent MIC_90_ of 7.81–1.95 µg/mL (Figure 2). In addition, *P. brasiliensis* was particularly sensitive to the tested compounds; indeed, all nitrofurans tested against *P. brasiliensis* had an MIC_90_ of between 31.25 and 0.48 µg/mL (Figure 2).

### 3.3. Determination of MIC_90_ and MFC for Candida species and Cryptococcus neoformans

For *Candida* species, the MIC_90_ ranged from 3.9 to values higher than 250 µg/mL (Table 1). Most compounds showed a fungicidal profile with MFC values from 7.81 to greater than 250 µg/mL (Table 1). For *Cryptococcus neoformans*, the MIC_90_ of nitrofuran derivatives also ranged from 3.9 to greater than 250 µg/mL, with a fully fungicidal profile, the MFC being equal to or two times greater than the MIC_90_ (Table 1). The susceptibility test results for the strain of *C. krusei* with AmB gave an MIC_90_ of 1 µg/mL, proving the quality of the experiments.

### 3.4. Determination of MIC and MFC for H. capsulatum and P. brasiliensis

The MIC_90_ scores ranged from 0.48 to over 250 µg/mL for the *H. capsulatum* strain and from 0.48 to 31.25 µg/mL for *P. brasiliensis* (Table 2). Compound **1** showed a fungistatic profile when tested against the *P. brasiliensis* strain; otherwise, the other active compounds showed a fungicidal profile for both fungi. AmB showed an MIC_90_ of 0.03 µg/mL for *H. capsultum* and 0.13 µg/mL for *P. brasiliensis*.

### 3.5. Determination of MIC and MFC for T. mentagrophytes

For both *T. rubrum* and *T. mentagrophytes* dermatophyte strains, the MIC_90_ ranged from 0.98 to greater than 250 µg/mL (Table 3). Compounds **3** and **4** showed a fungistatic profile on *T. mentagrophytes,* while **16** showed a fungistatic profile on *T. rubrum*. TRB showed a MIC_90_ of 0.03 µg/mL to the control strain *T. rubrum* ATCCMYA-4438, proving the quality of the experiments.

### 3.6. Cytotoxicity Assay by the Resazurin Method and Selectivity Index

The cytotoxicity assay was performed using the resazurin colorimetric method. After quantifying the viability, the compounds’ inhibition index of 50% of cell proliferation (IC_50_) was calculated. For the A549 cell line, the IC_50_ ranged from 8.58 to 250 µg/mL (Table 4). The selectivity index (SI) ranged from 0.03 to 64.10 for *Candida* species, 0.04 to 8.57 for *C. neoformans*, 0.05 to 59.28 for *H. capsulatum*, 0.42 to 481.66 for *P. brasiliensis,* and 0.05 to 255.10 for dermatophytes. For the MRC-5 cell line, the IC_50_ ranged from 1.56 to 250 µg/mL (Table 5). The SI ranged from 0.006 to 61.66 for *Candida* species, 0.006 to 10.19 for *C. neoformas*, 0.006 to 130.33 for *H. capsulatum*, 0.49 to 520.83 for *P. brasiliensis*, 0.006 to 232.65 for *T. rubrum*, and 0.006 to 116.92 for *T. mentagrophytes*.

### 3.7. Toxicity on C. elegans Model

According to the selectivity index results, compounds **1**, **3**, **5**, **11** and **12** were evaluated in vivo in *C. elegans*.

All compounds were tested on AU37 larvae, which showed a viability higher than 80%, even when treated with the highest concentration of the tested compounds (250 µg/mL). Compounds **1** and **12** caused less toxicity in the larvae, which maintained viability more significant than 94% at a concentration of 250 µg/mL (Figure 3). The larvae treated with compounds **3**, **5** and **11** showed a decrease in viability only at the highest concentration value (250 µg/mL) when compared to the control (*p* < 0.01; *p* < 0.05; *p* < 0.001, respectively) (Figure 3). The larvae images were obtained from the wells with the highest concentration of compounds (250 µg/mL); as can be observed, there are few dead larvae (rod-shaped), and most of them are sinusoidal (alive) (Figure 4).

## 4. Discussion

Fungal diseases can affect the oral mucosa, skin, nails, hair, lungs, brain, or several organs and tissues simultaneously [6,18,19,24,25,35]. Constant global changes and the advent of new pandemics and chronic diseases favor the diffusion of fungal pathogens such as *Candida*, *Cryptococcus*, *Aspergillus*, *Trichophyton,* and dimorphic fungi such as *H. capsulatum* and *P. brasiliensis* [1,2,26].

Fungal diseases are often neglected, and the rapid and accurate identification of fungi in clinical practice is limited. Besides this, since some infections require a long period of drug intervention, not counting the high recurrence rates, treatments with antifungals can be costly in addition to their toxicity [1,72,73]. For this reason, the search for new effective therapeutic alternatives for fungal infections is emerging, combined with research that minimizes the use of mammals.

We used cell cultures, which have become one of the most frequently applied techniques to replace or complement animal studies in vivo, helping in the toxicity and efficacy tests of new molecules [74,75]. In addition to cell cultures, non-mammalian alternative animals are widely used for pharmacological safety tests. *C. elegans* is classified as a nematode and measures about 1 mm in length, feeding on bacteria and fungi that decompose in the soil. It was characterized in the 1960s and has been essential in research, particularly in toxicology [76]. The AU37 strain has a glp-4 mutation that makes the larvae incapable of reproducing at 25 °C, while the mutation in sek-1 is responsible for making the larvae more sensitive to several pathogens [71,77].

One of the parameters used to verify the safety and potency of drugs or prototypes is the selectivity index. Values greater than ten are considered a sign of specificity and high selectivity [38,78,79]. Our findings show that nitrofuran **1** was the most promising among the tested compounds against *Candida* species, with MIC_90_ values ranging from 3.9 to 31.25 µg/mL and a selectivity index ranging from 7.69 to 30.79. Similar results with nitrofuran derivatives were obtained by De Vita et al. [45,47], with MIC_50_ values against *C. albicans* of between 0.5 and ≥128 µg/mL, and by Kamal et al. [46], with MIC values against *C. albicans* from 3.9 to 62.5 µg/mL. Furthermore, among the compounds tested on *C. elegans* larvae, compound **1** gave a viability percentage of 94% at a concentration of 250 µg/mL when we analyzed the acute toxicity of this compound, thus suggesting the low/null toxicity of this nitrofuran derivative.

Compound **5** showed higher potency against *C. neoformans* (MIC_90_ 7.81 µg/mL and MFC 15.62 µg/mL), with SI from 7.72 to 10.19 on A549 and MRC5 cells, respectively. When the acute toxicity in *C. elegans* larvae was analyzed, this compound gave a viability of 85.65% at the highest concentration tested (250 µg/mL). Reports of antifungal resistance in *Cryptococcus* strains are increasing [80,81]. However, fluconazole-resistant isolates have been reported [82,83] and their emergence is thought to be due to the frequent use of the drug as a preventive of cryptococcal meningitis or asymptomatic cryptococcal antigenemia [82,83,84].

The most active compound to *H. capsulatum* was **11** (MIC_90_ 0.48 µg/mL and MFC 0.48 µg/mL) with SI values of 25.50 and 130.33 on A549 and MRC5 cells, respectively. Otherwise, regarding *P. brasiliensis*, the most potent derivative was compound **3** (MIC_90_ 0.48 µg/mL and MFC 0.98 µg/mL), with SI values of 481.66 to 520.83 on A549 and MRC5 cells, respectively. Concerning acute toxicity in *C. elegans* larvae, compound **11** gave 80% viability, and compound **3**, 82.48% at the highest concentration tested (250 µg/mL). These fungi cause systemic mycosis, and their treatment is limited and with high levels of toxicity [30,34,85]. These data indicate that nitrofuran derivatives have good antifungal properties (both fungistatic and fungicidal) toward these species; on the other hand, they also showed a good selectivity and low toxicity, both on cell lines and on larvae.

Compound **12** was the most potent and the most selective on dermatophyte strains, showing good fungicidal and fungistatic activity against *T. rubrum* (MIC_90_ 0.98 µg/mL and MFC 1.95 µg/mL) and *T. mentagrophytes* (MIC_90_ 1.95 µg/mL and MFC 1.95 µg/mL). Furthermore, this compound showed low toxicity in vitro, with SI values toward *T. rubrum* of 255.10 on A549 and 232.65 on MRC5 cells, respectively, and SI values toward *T. mentagrophytes* of 128.20 on A549 and 116.92 on MRC5 cells, respectively. In *C. elegans*, the viability was 97.20% at the highest concentration tested (250 µg/mL). Dermatophytes are filamentous fungi with a high prevalence in the human population [38,62] and usually cause infections that are difficult to eradicate and often recur [38,39].

Our results showed that some nitrofuran derivatives have broad-spectrum antifungal activity, with low toxicity and great potential to treat infections caused by these fungi.

Overall, this study yielded hopeful results as some of the derivatives studied showed low toxicity in both in vitro and in vivo tests, and, for the most part, presented a fungicidal profile.

## 5. Conclusions

On the whole, this study prompted us to identify new nitrofuran derivatives with a potent and broad-spectrum antifungal activity that is mainly due to fungicidal action; naturally, the values of the MIC and MFC as determined varied in a wide range but this variation is to be expected, as there are different fungal strains and species. It is noteworthy that the most potent antifungal nitrofurans, compounds **1**, **3**, **5**, **11,** and **12,** showed low toxicity, both in vitro on A549 and MRC5 cell cultures, showing high SI values, and in vivo on *C. elegans* larvae, showing a viability higher than 80% at the highest concentration tested. These results showed that nitrofuran derivatives are promising compounds for treating fungal infections with a broad spectrum. New studies will be carried out to investigate their effectiveness in communities of microorganisms and in vivo in other alternative animals, to elucidate their mechanism of action.

## Data Availability

Not applicable.

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
