# Peer review of "Synthesis and Evaluation of the Antifungal and Toxicological Activity of Nitrofuran Derivatives"

_pharmaceutics, 2022, doi:10.3390/pharmaceutics14030593_

Round 1

Reviewer 1 Report

The work of Carolina Orlando Vaso et al. entitled “Synthesis and evaluation of the antifungal and toxicological activity of nitrofuran derivatives” presents interesting and valuable subject of study. The manuscript describes original results, presented logically and understandably. The pictures are prepared clearly. Supplementary information can be helpful in better understanding of the subject of study. I have two remarks, which of course do not diminish the value of manuscript.

Particular objections.

  1. The numbering of cited references should be improved – References [10, 50, 51, 52, 71] are not cited in the text of manuscript.
  2. The notation of Ref. [90] is incomplete.

In summary, in my opinion the paper, after above minor corrections, may be recommended for publication.

Author Response

As suggested, we checked the list of cited references and corrected the manuscript accordingly.

Reviewer 2 Report

The authors in their manuscript entitled as "Synthesis and evaluation of the antifungal and toxicological activity of nitrofuran derivatives" present a series of nitrofuran adducts with a broad-spectrum antifungal activity. This work exploits previously reported findings and presents low novelty. Even, the authors do not characherize the novel derivatives (elemental analysis or HRMS and 13C-NMR), which is minority.

Author Response

We have added the chemical experimental part in the main text with the spectroscopic and chemical-physical data of the new compounds and of those not fully described in the literature; consequently, we have added a discussion part of the spectroscopic data.

Reviewer 3 Report

The article realized by Vaso et al is very interesting in terms of the results of antifungal tests performed on 5-nitrofuran derivatives, in the context of research to discover new antimicrobial agents, in an attempt to overcome the resistance to known antifungals.

For a better presentation of the tested compounds, I recommend the authors to present them in a table, along with their main properties (molar masses, melting points). Compounds that do not have the original structure and that were obtained by reproducing some techniques from the literature, how do the authors comment on the yields? How were they purified?

The authors established as objectives of the paper both the synthesis and the testing of the biological action of 5-nitrofuran derivatives. However, the synthesis part is relegated to the background as supplementary material.

I recommend the authors to include the synthesis of compounds 2, 3, 15-17 in the article, completing with the following aspects, in order to characterize them and to demonstrate their synthesis:

- Melting temperatures (along with their recording method, equipment)

- IR absorption spectra

- 13C-NMR spectra

- Mass spectra

The obtained spectral data must be interpreted and discussed.

How did the authors determine the isomerism of compounds 15-17?

All spectra of compounds 2, 3, 15-17 must be presented as Supplementary Material.

I believe that if the authors complete the article with the suggested aspects, its level will increase so that it can be published.

Author Response

As suggested by th reviewer, we have added the chemical experimental part in the main text with the yields and the spectroscopic and chemical-physical data of the new compounds; consequently, we have also added a discussion of the spectroscopic data. The IR, 1H and 13C NMR spectra were added in the supplementary file. Compounds 1, 4-14, that don’t have original structure, were synthesized and purified according to the literature procedures, and they were obtained in similar yields as reported in the literature. As concerned the presentation of the new compounds, we would prefer not to add a further table as many are already present for the description of the biological results. Furthermore, the number of new compounds described is not such as to more conveniently require the presentation of their data in the form of a table.

Reviewer 4 Report

The manuscript entitled "Synthesis and evaluation of the antifungal and
toxicological activity of nitrofuran derivatives" reports the syntheses and antifungal activity of some nitrofuran derivatives.  The manuscript is decently written and the experiments appear to be competently designed and executed. I have the following suggestions:

  1. Some of the compounds (e.g. 1,4-14) are known in the literature, not "newer". The authors must therefore modify the article in depth, especially with regard to the chemistry of nitrofuran derivatives and their physical and spectrometric properties which must be compared with the existing data in the literature.
  2. As the above said compounds were already reported, therefore also provide CAS#, if available.
  3. Provide a scheme for the syntheses of nitrofuran derivatives.
  4. Attach some spectra in the supplementary file.
  5. Introduction too vague. Focus on the design of nitrofuran deriv. as antifungal agents.

Author Response

We have added the chemical experimental part in the main text with the yields and the spectroscopic and chemical-physical data of the new compounds; consequently, we have also added a discussion part of the spectroscopic data. The schemes of chemical synthesis were added. The IR, 1H and 13C NMR spectra were added in the supplementary file. A table with the CAS number of already known compounds was added in the supplementary file. As regards the introduction, we have tried to underline the importance of fungal infections and drug resistance problems by trying to cite the scarce literature available on the antifungal activity of nitrofurans.

Round 2

Reviewer 2 Report

The authors have provided the required data and their manuscript is suitable for publication.

Reviewer 3 Report

I consider that the scientific level of the article by Vaso and col. has increased through the additions related to the chemical part of the research. This was necessary because one of the objectives of the research was the synthesis of these nitrofuran derivatives. I consider that the article can be published in this form.